# How Has Climate Change Driven the Evolution of Rice Distribution in China?

**DOI:** 10.3390/ijerph192316297

**Published:** 2022-12-05

**Authors:** Guogang Wang, Shengnan Huang, Yongxiang Zhang, Sicheng Zhao, Chengji Han

**Affiliations:** 1Institute of Agricultural Economics and Development, Chinses Academy of Agricultural Sciences, Beijing 100089, China; 2School of Economics and Management, China Agricultural University, Beijing 100089, China; 3State Key Laboratory of Urban and Regional Ecology, Research Center for Eco-Environmental Sciences, China Academy of Sciences, Beijing 100089, China

**Keywords:** rice, GEE, crop distribution, semi-parametric quantile regression

## Abstract

Estimating the impact of climate change risks on rice distribution is one of the most important elements of climate risk management. This paper is based on the GEE (Google Earth Engine) platform and multi-source remote sensing data; the authors quantitatively extracted rice production distribution data in China from 1990 to 2019, analysed the evolution pattern of rice distribution and clusters and explored the driving effects between climatic and environmental conditions on the evolution of rice production distribution using the non-parametric quantile regression model. The results show that: The spatial variation of rice distribution is significant, mainly concentrated in the northeast, south and southwest regions of China; the distribution of rice in the northeast is expanding, while the distribution of rice in the south is extending northward, showing a spatial evolution trend of “north rising and south retreating”. The positive effect of precipitation on the spatial distribution of rice has a significant threshold. This shows that when precipitation is greater than 800 mm, there is a significant positive effect on the spatial distribution of rice production, and this effect will increase with precipitation increases. Climate change may lead to a continuous northward shift in the extent of rice production, especially extending to the northwest of China. This paper’s results will help implement more spatially targeted climate change adaptation measures for rice to cope with the changes in food production distribution caused by climate change.

## 1. Introduction

Climate change has become an indisputable fact and is already impacting many areas of ecological and social activities worldwide [1]. Agriculture is one of the most sensitive and vulnerable industries to climate change [2]. The cropping system, structure, distribution and production capacity will all be affected by climate factors [3,4,5,6]. As one of China’s major food crops, rice has a total annual consumption of around 200 million tonnes, with more than 85% used for primary food consumption, which is vital in China’s food production and security strategy [7]. In the last 20 years, the distribution of rice in China has withstood extraordinary changes [8]. On the one hand, the total area of rice sown has decreased in the south [9]. On the other hand, the regional distribution has changed, of which the area planted in the traditional southern rice growing areas fell rapidly and the area sown in the north, especially in the northeast, shows a significant increasing trend [10]. Climate change adds uncertainty to the long-term changes in the distribution of rice production [11].

At present, research on the effects of climate change has gone beyond climate science, with scholars from different domains exploring the impacts of climate change on the distribution of crops from different perspectives [12]. The literature has mainly used spatial autocorrelation [13], the regional centre of gravity analysis [14] and production functions [15]. The impact of climate change on grain crop production has both advantages and disadvantages. Nevertheless, the disadvantages outweigh the advantages overall, and different climate variables have different impacts on different crops and regions [16,17]. In recent years, the increased heat caused by climate change has been conducive to expanding the grain sown area and producing more grain [11]. Increasing rainfall and CO2 concentrations are beneficial for crop production to some extent, but high temperatures may negate this effect in some areas [18]. Similarly, climate change had a negative impact on grain production by expanding pest and disease occurrence areas, shortening crop growth cycles, and increasing the frequency of extreme weather events [19].

Since 1960, China’s 10-degree Celsius cumulative temperature contour has shifted significantly northwards [20]. Since the 1990s, the temperature in the northeast of China has increased dramatically, causing the rice-growing areas in the northeast to spread northwards, with areas previously unsuitable for rice growing, such as the Yichun and Heihe areas, now being able to grow rice [21]. In the analysis of the influence mechanism, works of literature mainly use statistical models and crop growth models [22,23]. The relationship between cumulative temperature, precipitation and other climatic factors and the distribution of rice cultivation has been analysed [24]. Climate change is believed to have an essential impact on rice yield, growth and distribution [25].

The current literature reveals the effects of climate on the evolution of rice distribution. However, the relatively few quantitative analyses of the relationship between climate change and the evolution of rice distribution are still limited to some regions, and analyses focusing on the large-scale changes in crop distribution from a national perspective are insufficient. Quantitative remote sensing is an important tool for studying and analysing changes in the distribution of crops at larger scales and over longer time series [26]. The development of the Google Earth Engine (GEE) cloud computing platform has laid the foundation for large-scale remote sensing data applications [27]. The GEE platform has been widely used for large-scale, quantitative remote sensing studies. The crop identification tools under the GEE platform have been developed relatively quickly. The crop identification tools under the GEE platform have been developed more rapidly. The development of remote sensing by big data analysis platforms and quantitative remote sensing technology has laid a great foundation for analysing crop distribution on a larger spatial scale.

Therefore, it is possible to explore the relationship between climate change and rice distribution conversion. We used the remote sensing panel data of rice distribution in China from 1990 to 2019, mapped the spatial distribution pattern of rice production in China and analysed its evolution mechanism by climate change. Further, we predicted the rice distribution in China in 2035 and 2050 based on different CMIP6 climate simulation scenarios. Under the conditions of climate change, rice, one of the most important food crops in China, is also one of the most important crops in the world, so clarifying the dynamic effects of climate change on rice distribution will help to identify and predict future changes in rice distribution.

## 2. Materials and Methods

### 2.1. Methodology

#### 2.1.1. GEE-Based Data Extraction for Rice Distribution

This paper uses the GEE platform to extract spatial data of rice distribution under a long time series. High-precision rice classification extraction includes two stages: selecting and pre-processing multi-source remote sensing data. The other is the calculation of key identification indicators based on the pre-processed data. Among them, the key identification indicators include spectral parameters, vegetation indices, topographic factors, etc. Then, the spatial distribution data of rice production were obtained by constructing an X-mean estimation model, combined with visual interpretation for reclassification [28]. We also consider the existence of cloudy and rainy climatic characteristics during the key phenological periods of rice growth in China, which cause difficulties for multispectral remote sensing monitoring. This paper uses the Top of Atmosphere Reflectance (TOA) generation algorithm of the phase-level Landsat series satellites to synthesise TOA images with very few clouds, and set a cloud score threshold to reconstruct cloud-free images for each of the key phenological periods [29]. To achieve high-precision identification and classification of rice production layout, this paper firstly constructs a dynamic normalised vegetation index NDVI [30]. Furthermore, we used the surface water body index (LSWI) to describe vegetation canopy and soil surface water content [31]. We extracted rice according to the difference between the vegetation index and other features in different regions during key phenological periods. Other indicators include surface temperature data based on the inversion of MODIS data products, and slope data calculated from DEM data based on the SRTM satellite, which are rectified into a time–space–spectrum multidimensional array for classification using the X-mean algorithm [32]. After the accuracy test, the overall classification accuracy of the rice distribution data extracted by the above algorithm was 79.43%, and the user accuracy was 98.79%, which could be sufficient for the demand for quantitative analysis of the changes in the rice distribution.

#### 2.1.2. Landscape Pattern Index

The Landscape Pattern Index (LPI) is a quantitative index that reflects landscape units’ internal compositions and spatial configurations [33]. This paper uses the landscape pattern index method to analyse rice’s spatial and temporal distribution characteristics. In selecting the landscape index, the patch area index (CA) was selected to reflect the scale of rice production in different counties of China, to reflect the landscape pattern characteristics of the spatial layout of rice production and to minimise information redundancy [34]. The specific calculation methods and meanings are as follows.(1)$$CA=\overset{a}{\underset{j=1}{\sum }}a_{ij}$$
where $a_{ij}$ denotes rice growing plots in counties, the patch area index can reflect the scale of the rice production system growing in each county.

#### 2.1.3. Centre of the Gravity Model

The concept of the centre of gravity refers to a point in regional space where the forces in all directions remain relatively balanced before and after the point [35]. In geography, the centre of gravity indicates that after dynamically weighing the magnitude of the forces acting between regions, the distribution moves in the direction of the greater force, and the direction of movement is the direction of change in the spatial pattern of the variable. Based on this, this paper uses the centre of the gravity model to analyse the spatial migration relationship of rice production in China. The centre of gravity model is calculated as follows.
(2)Xw¯=∑i=1nwixi∑i=1nwi
(3)Yw¯=∑i=1nwiyi∑i=1nwi
where $w_{i}$ is the area i the total number of factors, $\left(x_{i},y_{i}\right)$ are the coordinates of the geometric centre of the area i and $\left(\bar{X_{w}},\bar{Y_{w}}\right)$ are the coordinates of the centre of gravity of the element.

#### 2.1.4. Econometric Model

Ordinary least squares regression of means only reflects the structural relationship between data means, and its estimates may not be robust when the sample data do not satisfy the classical assumptions of traditional econometric models, so this paper uses quantile regression to overcome its averaging effect to give comprehensive information about the explanatory variables [36]. In order to eliminate the effect of the raw data in terms of magnitude, the explanatory variables were natural logarithms separate from the explanatory variables mentioned above, and their models were constructed as follows.
(4)$$Q_{y_{i|x_{i}}}\left(\tau |x_{i}\right)=\alpha +\beta_{1}lntemp_{ij}+\beta_{2}lnperp_{ij}+\beta_{ij}lnx_{ij}+u_{it}$$
where $Q_{y_{i|x_{i}}}\left(\tau |x_{i}\right)$ denotes the values of the patch area index for rice at different quartiles. The $temp_{ij}$ denotes the annual accumulation temperature level greater than 10 degrees Celsius in each county. The $perp_{ij}$ denotes the total annual precipitation in each county. The $x_{ij}$ denotes control variables, including land and socio-economic factors.

In order to analyse the relationship between climatic conditions on the distribution of rice, and to take into account the differences in climatic effects on rice production in different counties, this paper uses quantile regression to estimate the effects of climatic factors. Still, there may be highly non-linear correlations between the variables in each quantile of the quantile regression model. The majority of the previous studies are based on the linear hypothesis and employ a linear model to study the relationship between economic variables. However, different economic subjects are commonly interrelated, and the different economic variables representing each economic subject often have complicated relationships. In other words, there are many nonlinear relationships among economic variables [37], If a linear model is adopted to fit the association between economic variables without considering the nonlinear relationship between them, it will inevitably lead to problems such as poor robustness and biased parameter estimation. As a tool to study nonlinear relationships, the nonparametric regression model has many advantages over a traditional linear model. The nonparametric regression model is data-driven, meaning that the relationship between economic variables is completely dependent on the variable data [38]. It is difficult to fit such a complex model structure with a linear quantile model, so this paper adds a semi-parametric regression component to the quantile regression model [39]. The additive model estimates an additive approximation to the multivariate quantile regression equation by smoothing the additive term for each individual with a single variable, thus avoiding the “dimensionality curse” of traditional non-parametric models [40]. The additive quantile regression model has several advantages: (1) Semi-parametric linear regression models contain both linear and non-linear components, allowing a comprehensive examination of the linear and non-linear effects of various climatic conditions on rice distribution; (2) The parametric component of the model requires a smaller sample size than non-parametric regression models, circumventing the problem of traditional non-parametric models relying on large samples; (3) The assumptions of the non-parametric model are more relaxed than those of the parametric model, and the robustness of the non-parametric and semi-parametric models is significantly better than that of the parametric model for samples with non-normal distribution and outliers. 

The basic construct of the addable semi-parametric quantile regression model is as follows.
(5)$$Q_{y_{i|x_{i},z_{i}}}\left(\tau |x_{i},z_{i}\right)=x_{i}^{\prime }\beta +\overset{J}{\underset{j=1}{\sum }}g_{j}\left(z_{j}\right)$$
where τ(0<τ<1) is the quantile of the model; $x_{i}^{\prime }\beta$ is the parametric part of the model and $\overset{J}{\underset{j=1}{\sum }}g_{j}\left(z_{j}\right)$ is the non-parametric part of the model.

### 2.2. Variables

In order to analyse the influence of climatic and environmental conditions on the distribution of rice, one must consider the principles of relevance, objectivity, comparability and consistency in the selection of variables. This paper takes the effect of climatic and environmental conditions, such as accumulated temperature and precipitation, on the distribution of rice into consideration.

(1)Cumulative temperature. The literature shows that when the ambient temperature is low, a certain degree of warming can promote photosynthesis of the plant leaves, but when the temperature exceeds a certain threshold, photosynthesis of the leaves will be inhibited [41]. In the process of crop growing, the accumulation of plant carbohydrates is inseparable from photosynthesis. Maintaining a certain level of cumulative temperature can promote photosynthesis in plants, resulting in relatively more energy accumulation and, thus, higher yield expectations [42]. Still, when the annual cumulative temperature in the region is too high, plants’ growth rates and respiration levels are relatively higher, and the accumulation of their output may be inhibited. For this reason, an annual cumulative temperature greater than 10 degrees Celsius was chosen as the explanatory variable to characterise temperature differences between counties.(2)Precipitation. Change in precipitation is an important constraint on the development of food production [43]. It has been shown that the expected level of food production is positively related to the amount of water used in planting. This paper uses annual precipitation as the core explanatory variable to analyse the impact of changes in precipitation conditions on rice production.

In addition to climatic factors, rice production is also influenced and constrained by land and socio-economic conditions. Considering the availability of data and avoiding endogeneity and covariance between variables as far as possible, this paper selects soil organic matter content [44], land slope level [45], the average distance between rivers and arable land and elevation as control variables for land conditions. Additionally, we include the average distance between arable land with the roads and railways; we also consider the nighttime light index as a control variable for socio-economic factors.

(1)Land. Land is one of the most basic factors of crop production. The literature has shown that changes in the soil environment have a much higher impact on crop yield levels than changes in the climate environment [46]. The quality of arable land is a key factor in the level of food production, and the expected level of crop output will increase when the soil environment in which arable land is located is improved. As an important component of the soil, soil organic matter not only contains various nutrients for plant growth, but also regulates the physical and chemical properties of the soil, improves the microbiological environment and soil structure and enhances soil stability [47]. At the same time, the topographical features of arable land may significantly impact the distribution of different crops [48]. The most suitable slope range exists for the crops, and flat ground is conducive to achieving a large-scale and mechanised crop distribution, reducing the overall food production costs and improving food production efficiency, but it also has significant advantages in maintaining water and soil drainage and irrigation. Therefore, this paper selects soil organic matter content, land slope level, the average distance between arable land and rivers and elevation as land condition control variables to control the effect of land environmental conditions on the layout of rice production.(2)Socio-economic factors. Taking complete account of the heterogeneous impact of differences in socio-economic endowment characteristics across counties on the distribution of rice, this paper uses the average distance of land for rice from roads and railways and the nighttime light index as control variables to offset the impact of differences in locational endowments on the distribution of rice [49].

The relevant variables studied in this paper are explained in Table 1.

### 2.3. Data

#### 2.3.1. Data on Climatic Conditions

In this paper, the annual cumulative precipitation and annual temperature totals higher than 10 degrees Celsius are selected as the core representations of the two climate factors of temperature and precipitation. The daily average temperature and precipitation totals are extracted based on the daily observation data from more than 800 meteorological observation stations distributed throughout China by the National Meteorological Observatory of China. For the calculation of cumulative temperature, the average daily temperature higher than 10 degrees Celsius was screened and summed on an annual basis to obtain the annual cumulative temperature higher than 10 degrees Celsius; the annual sum of daily precipitation within each meteorological observation station was used to obtain the annual precipitation of that meteorological observation station. Subsequently, the latitude and longitude coordinates of the meteorological stations were imported into ArcGIS for matching, and the Kriging method was used for spatial interpolation to obtain a spatial raster of the annual precipitation and annual cumulative temperature. Finally, the geographic vector boundary data for each county was introduced in ArcGIS, and the raster data of annual temperature and precipitation were spatially averaged for each county through ArcGIS.

To analyse the future climate changes, this paper selects three future scenarios for 2035 and 2050 from the Shared Socioeconomic Pathways (SSPs) of CMIP6 data SSP1-2.6, SSP2-4.5 and SSP5-8.5, and selects three models for China with reference to the existing experimental results (Table A1).

#### 2.3.2. Soil Condition Data

In order to control the effect of land on the distribution of rice, soil organic matter content, land slope level, the average distance between arable land and rivers and elevation were selected as control variables. The soil organic matter content data were obtained from the spatial data of soil type distribution in China from the Institute of Geographical Sciences and Resources, Chinese Academy of Sciences. Based on the available literature and combined with the data of the second national soil census in China, the organic matter content of different soil types was calculated separately according to their distribution, and the calculation criteria were as follows (Table A2).

The data on the land slope, elevation and rivers are obtained from the Resource and Environment Science and Data Centre of the Chinese Academy of Sciences. In this paper, all the data are taken at the basic scale of counties, and the spatial mean values of each county and district are extracted separately as the basic unit.

#### 2.3.3. Socio-Economic Data

In this paper, the average distance between arable land with roads and railways and the night-time light index of each county are used as control variables. The above socio-economic variables are obtained from the Resource and Environment Science and Data Centre of the Chinese Academy of Sciences. In contrast, the vector data of administrative divisions of Chinese sub-counties are obtained from the National Basic Geographic Information Database of the State Bureau of Surveying and Mapping. In the process of data analysis, data with significant discrepancies were eliminated, and county units with abnormal values or missing data in the year are replaced with data from neighbouring years.

## 3. Results

### 3.1. Spatial Pattern and Evolution of Rice Production in China

The X-mean algorithm was used to estimate the spatial distribution of rice in China from 1990 to 2019 by the GEE platform (Figure 1). The regional distribution of rice in China is mainly located in the south, but this spatial expansion continues to the north, showing a trend of “north rising and south retreating”. The centre of gravity of rice in 1990, 2000, 2015 and 2019 was calculated and plotted (Figure 2). The results show that between 1990 and 2019, the centre of gravity of rice production in China moved steadily in the northeast direction, from Luxi County, Xiangxi Tujia and Miao Autonomous Prefecture, Hunan Province, reaching to Suizhou County, Suizhou City, Hubei Province, in 2019, involving four counties and cities in two provinces, moving a straight-line distance of 518.2 km. It shows that the scale of rice production in northern China grew significantly faster than in southern China between 1990 and 2019.

To further analyse the spatial distribution changes of rice in China, this paper calculates the values of each landscape pattern index of the county to quantify the distribution characteristics of rice production in each county in China. From the patch area index (Figure 3), the traditionally dominant areas of rice production in China are mainly concentrated in the Sichuan Basin, the Yunnan–Guizhou region, the middle and lower reaches of the Yangtze River Plain, etc. Over time, the distribution of rice in the northeast has been expanding, gradually spreading from the northeast of the Northeast Plain to cover the entire northeastern areas, while the distribution of rice in southern regions has also been expanding northward. Still, rice in Guangdong, Fujian and other southeastern coastal regions has been reduced. It could be caused by these areas being economically developed, the land distribution being fragmented and hilly, the opportunity cost of rice production being relatively high and a common tendency for farmers to de-farm and de-grain their land.

To further reveal the dynamic process of the spatial pattern of rice in China, the study area was divided into low (−1.5 to −0.5 standard deviations), medium (−2.5 to −1.5 standard deviations)and fast (<−2.5 standard deviations) decreasing zones, stable (±0.5 standard deviations) zones, and low (0.5 to 1.5 standard deviations), medium (1.5 to 2.5 standard deviations) and fast (>2.5 standard deviations) increasing zones according to the magnitude of change in the rice patch area index from 1990 to 2019 (Figure 3 left). At the county scale, the decreasing zones of rice production scale in China’s counties from 1990 to 2019 are much smaller than the increasing zones. Specifically, the increasing zones are concentrated in the Northeast Plain and the southern part of the North China Plain; the decreasing zones are concentrated in the Sichuan, Guangdong, Fujian and Yunnan provinces. Statistically, the rice sown area in these four provinces decreased by 39.15%, 43.60%, 59.02% and 17.21%, respectively, between 1990 and 2019.

### 3.2. Climate Drivers of the Evolution of Rice Distribution in China

#### 3.2.1. Panel Quantile Model Applicability Test Results

Before parameter estimation, the variables were tested for smoothness, and the ADF–Fisher test was used to weigh the advantages and disadvantages of the smoothness testing methods. The test results (Table 2) showed that the patch area index, an annual cumulative temperature higher than 10 degrees Celsius and annual precipitation from 1990 to 2019 were all first-order differential smooth series, which satisfied the conditions for applying the panel quantile model controlling for time effects.

#### 3.2.2. Linear Influence of Climatic Factors on the Rice Distribution

This paper takes over 2000 counties in China from 1990 to 2019 as the research scale unit, estimates the climate effect of rice distribution in each county and selects three quartiles of 0.25, 0.5 and 0.75, respectively. The impact of climate on the rise distribution in different counties was investigated according to the different sub-quartiles.

Table 3 shows various factors influencing rice distribution in China at different quantile points. In terms of climatic factors, climate conditions have a significant influence on the scale of rice production in China, as shown by the following: (1) The annual accumulation temperature has a significant negative influence on the scale of rice production, i.e., as the quantile point rises, the less negative influence the accumulation temperature level has on the scale of rice production; (2) Precipitation has a significant positive influence on the patch area index of rice at both the 0.5 quantile point and the 0.75 quantile point, and as the effect of this relationship decreases as the quantile rises, i.e., in counties with small-scale rice production, the distribution of rice is more sensitive to changes in precipitation.

In terms of land conditions, there is a significant negative relationship between the scale of rice production and soil organic matter content. The influence of land factors on the scale of rice production decreases as the quantile rises. As the quantile point rises, the scale of rice production expands, and the negative effect of soil organic matter content on the scale of rice production decreases.

In terms of socio-economic factors, the average distance between arable land with roads and railways has a significant positive effect on rice production, and its effect tends to weaken as the quantile rises; the level of economic development, as characterised by the nighttime light index, has a significant adverse effect on the scale of rice production, with the more economically developed counties having a relatively smaller scale of rice production. As the scale of rice production rises, the economy’s negative effect on the scale of rice production gradually decreases. The negative impact of the economy on the scale of rice production gradually decreases as the scale of rice production rises.

#### 3.2.3. Non-Linear Driving Effects of Climatic Factors on Rice Production

To further investigate the relationship between the influence of climatic factors on the rice distribution, after controlling for other variables as linear influence relationships, this paper estimated the effect of three different quartiles for 0.25, 0.5 and 0.75 on the level of accumulated temperature and precipitation. There was a significant non-linear relationship between climatic factors such as accumulated temperature and precipitation on the distribution characteristics of rice (Table 4).

The effect of temperature on the spatial distribution of rice was significant (Figure 4), and it is approximately in the models at different quartiles. The analysis revealed (Table 4) that the positive effect of cumulative temperature on rice distribution was characterised by a significant ‘double peak’, i.e., there were two effective temperature ranges that could promote rice production expansion, with the effective cumulative temperature ranges for the 0.25 quantile at 2782.935–2918.037 °C and 5403.880–6229.566 °C, the effective temperature ranges for the 0.5 quantile are 2531.201–3444.651 °C and 5403.88–5941.196 °C and the effective cumulative temperature ranges for the 0.75 quantile are 3202.229–3611.886 °C and 5403.880–5802.05°C. The above fields correspond to the main distribution ranges of single-season rice in the northern region and double-season rice in the southern part of China. The positive effect on the area under rice cultivation is most significant in the ranges of 2800 to 3600 °C and 5400 to 6200 °C. Combined with the actual distribution of rice cultivation in China, when the accumulated temperature is below about 2800 degrees Celsius, the temperature will become the main stress factor for rice production and cultivation, which will limit the growth of rice and the growth of cultivation scale.

The effect of precipitation on the spatial distribution of rice was significant (Figure 5), and it is approximately in the models at different quartiles. Additionally, the effect of precipitation on the rice distribution shows a significant non-linear dynamic relationship within the range of values. When the value of precipitation is in a high range, it has a significant positive effect on the rice distribution. Further analysis revealed (Table 5) that the positive effect of precipitation on the spatial distribution of rice has significant interval characteristics, i.e., the annual precipitation has a significant positive effect on the distribution of rice only when it is above a certain threshold, specifically, the effective precipitation range for the 0.25 quantile is 870.559–1670.085 mm, the effective precipitation range for the 0.5 quantile is 849.009–1670.085 mm and the effective precipitation range for the 0.75 quantile is 849.009–1712.451 mm. The results show that the content of annual precipitation around 800–1700 mm significantly positively affects the area under rice cultivation. The estimation results indicate that precipitation, as an important variable affecting soil water content and rice growth, can only achieve a facilitating effect on the expansion of rice production layout in a relatively high interval range. When precipitation is in a relatively low interval range, precipitation may inhibit rice cultivation. Water resource constraints are an important variable affecting the spatial layout of rice.

### 3.3. Prediction of the Evolution of the Rice Distribution in China

Further, this paper estimates the changes in rice distribution between 2035 and 2050 through three future scenarios in CMIP6, SSP1-2.6, SSP2-4.5 and SSP5-8.5, assuming no changes in land conditions and socio-economic factors (Figure 6). The predictions show that climate change brings higher accumulated temperature and abundant precipitation to the North China Plain, the Northeast Plain and along the Hexi Corridor in China, with the 800 mm precipitation line showing a northward shift. Under these conditions, the spatial distribution range of rice will shift further north. Specifically, the distribution range of double-season rice in the south will expand northwards, and the scale of rice cultivation in dry rice staggered areas such as Henan and Anhui may rise further. The suitable distribution range of single-season rice in the north will extend westwards, and the scale of rice production in Longnan, Baiyin, Zhangye and Qingyang in Gansu province will expand further.

## 4. Conclusions and Policy Implications

This paper uses the GEE platform to extract the rice distribution in China through multi-temporal multi-source remote sensing data and analyse the evolution characteristics of rice distribution in China between 1990 and 2019 using the landscape pattern index and the centre of gravity model. The study found that the spatial distribution of rice in China is significantly concentrated in the northeast, Sichuan Basin, Dongting Lake and Poyang Lake basin areas. The rice distribution focuses on the northeastern and southern parts of China. In contrast, rice cultivation continues to shrink in Sichuan, Guangdong, Fujian and Yunnan.

Based on a semi-parametric quantile regression model, we found a significant non-linear relationship between two major climatic factors, cumulative temperature and precipitation, on the evolution of rice distribution. The positive effect of precipitation on the spatial distribution of rice has a significant “bimodal” characteristic, i.e., there are two effective ranges of temperature that can promote the expansion of rice production. The positive effect of precipitation on the spatial layout of rice production has a significant threshold, and the effect will increase with the increasing amount of precipitation.

China’s current rice production is mainly concentrated in the middle and lower reaches of the Yangtze River, the Sichuan Basin, the Three Rivers Plain, etc. The above areas are located in the range of 800–1700 mm precipitation with abundant rainfall. But with changes in climatic conditions, the 800–1700 mm precipitation line in China will show a significant tendency to move northwards. At that time, the scale of rice cultivation in the southern North China Plain and along the Hexi Corridor may further be increased. Due to climate change, climatic conditions in traditional rice farming areas may continue to change, and rice production in conventional rice farming areas may be continuously affected. In contrast, the climatic environment in non-traditional rice farming areas may become suitable for rice production.

Therefore, policies to respond to climate change-driven changes in food production patterns should be actively developed, and different food security strategies should be developed opportunistically in different regions based on their climatic conditions and level of socio-economic development. From a global perspective, there is currently a trend of shifting rice distribution from south to north in China through the expansion of rice in the north, and the shrinking along the southeast coast. On the other hand, the expansion of rice production in the north, especially in the northeast, is caused by the changes in climate conditions. Therefore, in light of the difficult food security supply situation, inter-regional food production should be developed for local conditions while ensuring a certain rate of food self-sufficiency and security of the primary food supply. A moderate scale operation should be promoted in some areas where rice distribution is more concentrated.

## Figures and Tables

**Figure 1 ijerph-19-16297-f001:**
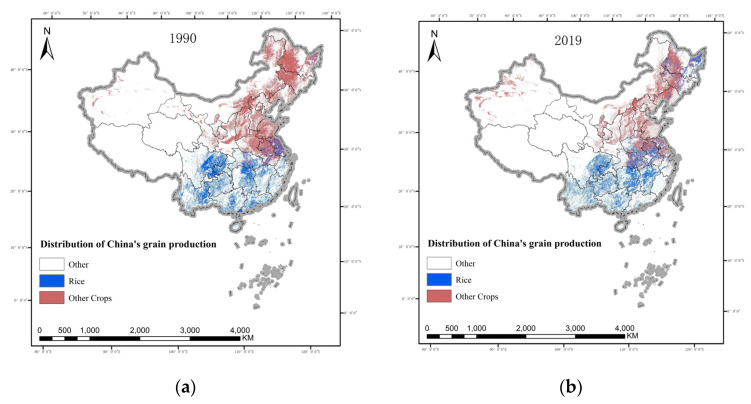
Distribution of grain production in China from (**a**) 1990 to (**b**) 2019.

**Figure 2 ijerph-19-16297-f002:**
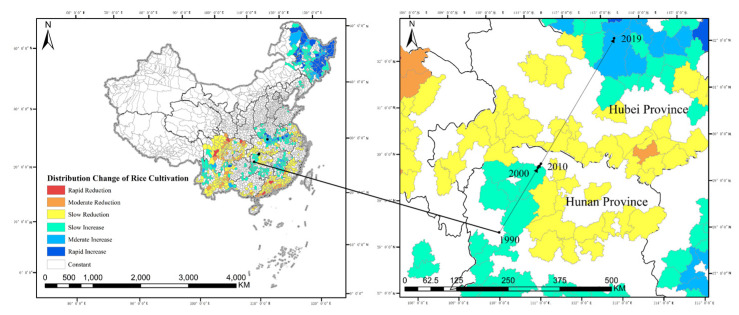
Change of patch area index and centre of gravity of rice production in China from 1990 to 2019.

**Figure 3 ijerph-19-16297-f003:**
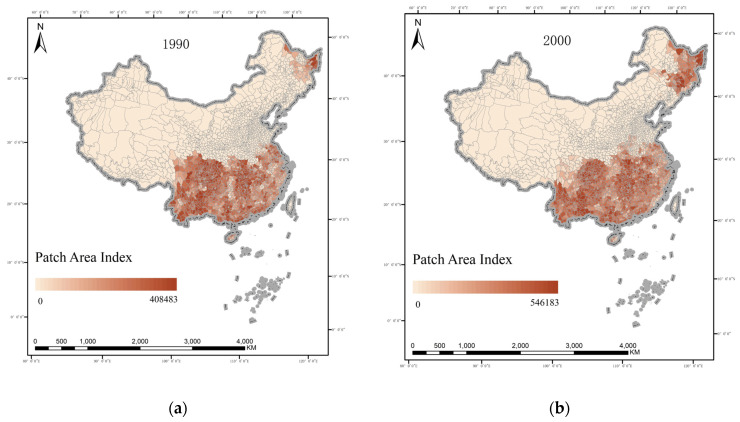
Change in area index of rice production patches in China in (**a**) 1990, (**b**) 2000, (**c**) 2010 and (**d**) 2019.

**Figure 4 ijerph-19-16297-f004:**
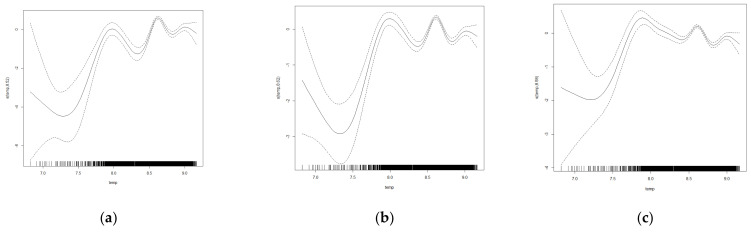
Results of non-linear quantile estimates of temperature on the rice patch area in (**a**) 0.25, (**b**) 0.5 and (**c**) 0.75 quantile.

**Figure 5 ijerph-19-16297-f005:**
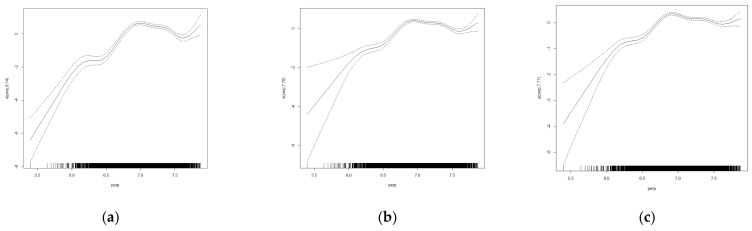
Results of non-linear quantile estimates of precipitation on the rice patch area in (**a**) 0.25, (**b**) 0.5 and (**c**) 0.75 quantile.

**Figure 6 ijerph-19-16297-f006:**
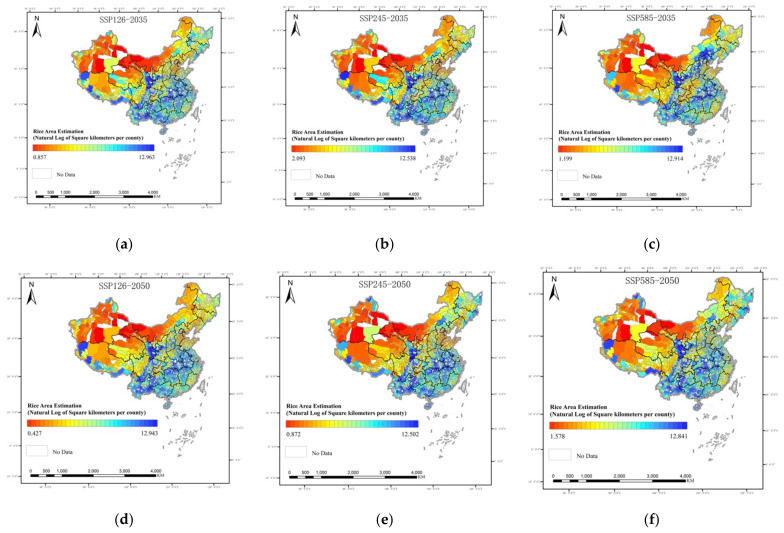
Spatial pattern of rice in China under (**a**) SSP126 in 2035, (**b**) SSP245 in 2035, (**c**) SSP 585 in 2035, (**d**) SSP126 in 2050, (**e**) SSP245 in 2050 and (**f**) SSP585 in 2050 scenarios.

**Table 1 ijerph-19-16297-t001:** Description of related variables.

	Variable	Description
Explained variables	Patch area index for rice	Scale of the rice production system growing in each county
Core explanatory variables	Annual cumulative temperature	The cumulative temperatures with greater than 10 degrees Celsius
	Annual precipitation	The annual cumulative precipitation
Control variables	Soil organic matter	Organic matter as a percentage of dry soil weight
	River distance	The average distance of arable land to river for each county
	Slope of the land	The average slope of arable land for each county
	Elevation	The average DEM for each county
	Road distance	The average distance of arable land to road for each county
	Rail distance	The average distance of arable land to rail for each county
	Nighttime Lighting Index	The index of Nighttime lighting

**Table 2 ijerph-19-16297-t002:** ADF test for key variables.

Variables	ADF	P
Patch Area index	−12.067	0.01
Higher than 10 degrees Celsius annual cumulative temperature	−13.771	0.01
Annual precipitation	−12.76	0.01

**Table 3 ijerph-19-16297-t003:** Panel quantile regression results of rice distribution.

Explanatory Variables	(1)			(2)			(3)			(4)		
	25th	50th	75th	25th	50th	75th	25th	50th	75th	25th	50th	75th
Constant term	6.926 ***	7.515 ***	7.895 ***	6.629 ***	7.895 ***	8.513 ***	6.764 ***	7.418 ***	7.828 ***	6.909 ***	8.055 ***	8.679 ***
	(0.073)	(0.057)	(0.054)	(0.256)	(0.200)	(0.205)	(0.078)	(0.052)	(0.054)	(0.318)	(0.263)	(0.242)
Annual cumulative temperature	−0.347 ***	−0.074	−0.028	−0.184 ***	−0.039	0.029 ***	−0.452 ***	−0.168 ***	−0.108 ***	−0.287 ***	−0.112 **	−0.061
	(0.071)	(0.052)	(0.054)	(0.069)	(0.052)	(0.053)	(0.070)	(0.047)	(0.050)	(0.071)	(0.049)	(0.052)
Annual precipitation	0.872 ***	0.515 ***	0.435 ***	0.551 ***	0.348	0.242 ***	0.791 ***	0.429 ***	0.337 ***	0.586 ***	0.356 ***	0.275 ***
	(0.084)	(0.061)	(0.065)	(0.082)	(0.061)	(0.064)	(0.083)	(0.057)	(0.062)	(0.086)	(0.057)	(0.062)
Soil organic matter				−1.850 ***	−1.443 ***	−1.171 ***				−1.897 ***	−1.435 ***	−1.184 ***
				(0.083)	(0.074)	(0.075)				(0.071)	(0.066)	(0.063)
River distance				0.079 ***	0.068 ***	0.067 ***				0.045 ***	0.037 ***	0.038 ***
				(0.004)	(0.003)	(0.003)				(0.005)	(0.003)	(0.003)
Slope of the land				0.212 ***	0.122 ***	0.079 ***				0.181 ***	0.100 ***	0.057 ***
				(0.019)	(0.015)	(0.015)				(0.025)	(0.021)	(0.020)
Elevation				0.116 ***	0.118 ***	0.112 ***				0.047 ***	0.047 ***	0.045 ***
				(0.007)	(0.006)	(0.006)				(0.010)	(0.009)	(0.009)
Road distance							0.078 ***	0.077 ***	0.074 ***	0.072 ***	0.072 ***	0.069 ***
							(0.006)	(0.004)	(0.004)	(0.006)	(0.005)	(0.004)
Rail distance							0.114 ***	0.095 ***	0.089 ***	0.092 ***	0.076 ***	0.071 ***
							(0.007)	(0.005)	(0.005)	(0.007)	(0.005)	(0.005)
Nighttime Lighting Index							−0.036 **	0.006	0.032 **	−0.098 ***	−0.073 ***	−0.050 ***
							(0.019)	(0.013)	(0.014)	(0.020)	(0.017)	(0.016)

Note: Superscripts *** and ** indicate statistical significance at 1%, 5% and 10% respectively; the standard error in parentheses; the 25th, 50th and 75th indicate represents the 0.25, 0.5 and 0.75 quartiles of the explanatory variables, respectively; model (1) estimates the effect of climate on rice distribution, model (2) estimates the effect of climate under controlled the land conditions, model (3) estimates the effect of climate under controlled socio-economic conditions, model (4) estimates the effect of climate under both controlled land and socio-economic conditions.

**Table 4 ijerph-19-16297-t004:** Results of semi-parametric quantile regression for rice production.

Explanatory Variables	(1)			(2)			(3)		
	25th	50th	75th	25th	50th	75th	25th	50th	75th
Constant term	10.613 ***	11.607 ***	12.463 ***	5.477 ***	7.635 ***	9.543 ***	6.018 ***	11.414 ***	15.025 ***
	(0.414)	(0.286)	(0.249)	(0.854)	(0.550)	(0.466)	(1.376)	(0.974)	(0.724)
Annual cumulative temperature							0.488 ***	0.021	−0.266 ***
							(0.147)	(0.107)	(0.079)
Annual precipitation				0.681 ***	0.501 ***	0.335 ***			
				(0.124)	(0.075)	(0.061)			
Soil organic matter	−1.346 ***	−1.014 ***	−0.744 ***	−1.615 ***	−1.246 ***	−0.956 ***	−1.309 ***	−0.820 ***	−0.486 ***
	(0.144)	(0.102)	(0.084)	(0.161)	(0.101)	(0.086)	(0.142)	(0.101)	(0.077)
River distance	0.046 ***	0.036 ***	0.033 ***	0.048 ***	0.038 ***	0.036 ***	0.044 ***	0.039 ***	0.036 ***
	(0.006)	(0.004)	(0.003)	(0.007)	(0.004)	(0.003)	(0.006)	(0.004)	(0.003)
Slope of the land	−0.051	−0.083 ***	−0.109 ***	−0.012	−0.032	−0.045 **	−0.019	−0.098 ***	−0.154 ***
	(0.034)	(0.024)	(0.020)	(0.039)	(0.024)	(0.020)	(0.034)	(0.024)	(0.019)
Elevation	0.117 ***	0.090 ***	0.076 ***	0.142 ***	0.103 ***	0.076 ***	0.093 ***	0.078 ***	0.069 ***
	(0.016)	(0.011)	(0.008)	(0.017)	(0.011)	(0.009)	(0.016)	(0.011)	(0.008)
Road distance	0.053 ***	0.061 ***	0.051 ***	0.056 ***	0.065 ***	0.052 ***	0.055 ***	0.061 ***	0.051 ***
	(0.008)	(0.006)	(0.004)	(0.009)	(0.006)	(0.005)	(0.008)	(0.006)	(0.004)
Rail distance	0.047 ***	0.049 ***	0.042 ***	0.060 ***	0.054 ***	0.045 ***	0.050 ***	0.047 ***	0.041 ***
	(0.009)	(0.006)	(0.005)	(0.010)	(0.006)	(0.005)	(0.008)	(0.007)	(0.005)
Nighttime Lighting Index	−0.056 **	−0.064 ***	−0.073 ***	−0.111 ***	−0.097 ***	−0.087 ***	−0.059 **	−0.061 ***	−0.076 ***
	(0.029)	(0.023)	(0.022)	(0.033)	(0.023)	(0.023)	(0.030)	(0.023)	(0.021)
s (annual cumulative temperature)	8.254 ***	8.510 ***	8.550 ***	8.524 ***	8.623 ***	8.583 ***			
s (annual precipitation)	7.697 ***	8.066 ***	7.879 ***				8.143 ***	7.764 ***	7.768 ***
R2	0.263	0.248	0.221	0.217	0.223	0.179	0.192	0.221	0.179
Deviance explained	22.70%	28.60%	48.30%	25.10%	19.60%	46.40%	25.90%	19.90%	46.50%

Note: Superscripts *** and ** indicate statistical significance at 1%, 5% and 10% respectively; the standard error in parentheses; the 25th, 50th and 75th indicate represents the 0.25, 0.5 and 0.75 quartiles of the explanatory variables respectively; model (1) estimates the non-linear effect of both annual cumulative temperature and annual precipitation, models (2) and (3) estimate the non-linear effect of annual cumulative temperature and annual precipitation, respectively.

**Table 5 ijerph-19-16297-t005:** Effective range of climatic factors contributing to the distribution of rice.

Climate Factors	25th		50th		75th	
	Effective Range	Impact Effects	Effective Range	Impact Effects	Effective Range	Impact Effects
Temperature accumulation	2782.935–2918.037	0.321–0.341	2531.201–3444.651	0.195–0.309	3202.229–3611.886	0.170–0.818
	5403.880–6229.566	0.202–0.398	5403.88–5941.196	0.054–0.063	5403.880–5802.056	0.066–0.100
Precipitation	870.559–1670.085	0.128–0.620	849.009–1670.085	0.125–0.432	849.009–1712.451	0.082–0.362

## Data Availability

Not applicable.

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
