# Peer review of "How Has Climate Change Driven the Evolution of Rice Distribution in China?"

_ijerph, 2022, doi:10.3390/ijerph192316297_

Round 1

Reviewer 1 Report

Based on the GEE platform and multi-source remote sensing data, authors quantitatively extracted rice production distribution data in China from 1990-2019. They analysed the evolution pattern of rice distribution and cluster. They used a non-parametric quantile regression model to explore the driving effects between climatic and environmental conditions on the evolution of rice production distributions. This shows that when precipitation is greater than 800 mm, there is a significant positive effect on the spatial distribution of rice production, and this effect will increase with precipitation increases. Climate change may lead to a continuous northward shift in the extent of rice production, significantly extending to the Northwest of China. The manuscript leaves a very good impression. The research topic is very relevant. At the same time, for some reason, there are few publications on this topic. This further enhances the importance of this work.

Line-11 “Based on GGE platform …” what does GGE mean. Please write it without abbreviation.

Line-14 change driving effect… to driving effects…..

Line-15 Before results one sentence regarding materials and methods should be added.  

Lines 59-60 “In the analysis of the influence mechanism, works         of literature mainly use statistical models and crop growth models” should be “In the analysis of the influence mechanism, works literature mainly uses statistical and crop growth models”.

Line 123 “…to minimise information redundancy” change to “…..to minimize information redundancy”

Lines 202-203 how cumulative temperature of each region was determined?

Line 205, why temperature greater than 10 degrees Celsius, was selected as the explanatory variable.

Lines 209-210 how was annual precipitation used? Do you use the precipitation data for all 12 months? If yes, then how were mean values calculated- Please justify!!

Table 5 & Table 6 what does caption *** indicates?

Table 5 & Table 6 need more justification- It would be better to justify these Tables with the significant positive correlation and significant negative correlation- Moreover, a clear relationship should be explained between all explanatory variables and (1), (2), (3) and etc. .Digits-

Author Response

Comment 1

Line-11 “Based on GGE platform …” what does GGE mean. Please write it without abbreviation.

Response:

We thank the reviewer for pointing this out. We have modified this mistake in the previous version of the article.

Comment 2

Line-14 change driving effect… to driving effects…..

Response:

We thank the reviewer for pointing this out. We have modified this mistake in the previous version of the article.

Comment 3

Line-15 Before results one sentence regarding materials and methods should be added.  

Response:

We thank the reviewer for pointing this out. We have modified this mistake in the previous version of the article as follows:

‘This paper is based on the GEE (Google Earth Engine) platform and multi-source remote sensing data, quantitatively extracted rice production distribution data in China from 1990 to 2019, analysed the evolution pattern of rice distribution and cluster, and explores the driving effects between climatic and environmental conditions on the evolution of rice production distributions by the non-parametric quantile regression model.’

Comment 4

Lines 59-60 “In the analysis of the influence mechanism, works of literature mainly use statistical models and crop growth models” should be “In the analysis of the influence mechanism, works literature mainly uses statistical and crop growth models”.

Line 123 “…to minimise information redundancy” change to “…..to minimize information redundancy”

Response:

We thank the reviewer for pointing this out. We have modified these mistakes in the previous version of the article.

Comment 5

Lines 202-203 how cumulative temperature of each region was determined?

Line 205, why temperature greater than 10 degrees Celsius, was selected as the explanatory variable.

Lines 209-210 how was annual precipitation used? Do you use the precipitation data for all 12 months? If yes, then how were mean values calculated- Please justify!!

Response:

We thank the reviewer for this insightful question. In line 248-258, we presented the method of how cumulative temperature and precipitation of each region was determined as follows:

‘For the calculation of cumulative temperature, the average daily temperature higher than 10 degrees Celsius was screened and summed on an annual basis to obtain the annual cumulative temperature higher than 10 degrees celsius; the annual sum of daily precipitation within each meteorological observation station was used to obtain the annual precipitation of that meteorological observation station. Subsequently, the latitude and longitude coordinates of the meteorological stations were imported into ArcGIS for matching, and the Kriging method was used for spatial interpolation to obtain a spatial raster of the annual precipitation and annual cumulus temperature. Finally, the geographic vector boundary data for each county was introduced in ArcGIS, and the raster data of annual temperature and precipitation were spatially averaged for each county through ArcGIS.’

In addition, according to previous literature[1-3], rice originated in tropical or subtropical regions and is a crop sensitive to low temperatures; below 10 degrees Celsius, crop growth and development can be severely disrupted, so we calculate annual cumulative temperatures at temperatures greater than 10 degrees Celsius.

Comment 6

Table 5 & Table 6 what does caption *** indicates?

Table 5 & Table 6 need more justification- It would be better to justify these Tables with the significant positive correlation and significant negative correlation- Moreover, a clear relationship should be explained between all explanatory variables and (1), (2), (3) and etc. .Digits-

Response:

We thank the reviewer for pointing this out. During the revision process, the authors have checked and revised all captions of figures and tables. We believe the paper's figures and tables read more clearly now.

  1. Krishnan, P.; Ramakrishnan, B.; Reddy, K.R.; Reddy, V.R. High-Temperature Effects on Rice Growth, Yield, and Grain Quality. In Advances in Agronomy; Elsevier, 2011; Vol. 111, pp. 87–206 ISBN 978-0-12-387689-8.
  2. Tollenaar, M.; Daynard, T.B.; Hunter, R.B. Effect of Temperature on Rate of Leaf Appearance and Flowering Date in Maize 1. Crop Sci. 1979, 19, 363–366, doi:10.2135/cropsci1979.0011183X001900030022x.
  3. Asseng, S.; Foster, I.; Turner, N.C. The Impact of Temperature Variability on Wheat Yields: IMPACT OF TEMPERATURE VARIABILITY ON WHEAT YIELDS. Global Change Biology 2011, 17, 997–1012, doi:10.1111/j.1365-2486.2010.02262.x.

Reviewer 2 Report

Dear Authors,

I have carefully reviewed the submission about the rice production distribution change in China from 1990 to 2019. The data were presented in detailed tables and figures.  The results will contribute to managing food production distribution caused by climate change. Besides, there is some part need to revise. I indicated my other comments in the text. in my opinion, the article should be worthy published after minor revision.

Author Response

We thank the reviewer for their valuable suggestions. 

We have modified these mistakes in the previous version of the article.

Reviewer 3 Report

Detailed notes on the manuscript are as follows:

1) Adapt the tables to the MDPI editorial requirements

2) Improve legibility of drawings (font)

3) Tables 5 and 6; what is "***" - homogeneous groups, significant at the p <... level?

4) Chapter 4 of 'Conclusions' has the form of a summary, I suggest changing the title to "Summary"

5) Lines 87-89 "The structure of this paper is as follows. Section 2 includes an introduction to the materials and methods we used. Section 3 describes the result of this study, including estimated result and predictions. Section 4 presents the conclusion of this study ”- are unnecessary, put the purpose of the work in this place.

Author Response

Comment 1

Adapt the tables to the MDPI editorial requirements;

Tables 5 and 6; what is "***" - homogeneous groups, significant at the p <... level?

Response:

We thank the reviewer for pointing this out. During the revision process, the authors have checked and revised all captions of figures and tables. We believe the paper's figures and tables read more clearly now.

Comment 2

Improve legibility of drawings (font)

Response:

We thank the reviewer for pointing this out. We have modified these mistakes in the previous version of the article.

Comment 3

Chapter 4 of 'Conclusions' has the form of a summary, I suggest changing the title to "Summary"

Response:

We thank the reviewer for their valuable suggestions. We agree that a revise is necessary for this article.

Comment 4

Lines 87-89 "The structure of this paper is as follows. Section 2 includes an introduction to the materials and methods we used. Section 3 describes the result of this study, including estimated result and predictions. Section 4 presents the conclusion of this study ”- are unnecessary, put the purpose of the work in this place.

Response:

We thank the reviewer for pointing this out. We have modified the introduction in the previous version of the article.
